# Anti-Adhesive Resorbable Indomethacin/Bupivacaine-Eluting Nanofibers for Tendon Rupture Repair: In Vitro and In Vivo Studies

**DOI:** 10.3390/ijms242216235

**Published:** 2023-11-12

**Authors:** Yi-Hsun Yu, Chen-Hung Lee, Yung-Heng Hsu, Ying-Chao Chou, Ping-Chun Yu, Chao-Tsai Huang, Shih-Jung Liu

**Affiliations:** 1Department of Orthopedic Surgery, Bone and Joint Research Center, Chang Gung Memorial Hospital-Linkou, Taoyuan 33305, Taiwan; m7048@cgmh.org.tw (Y.-H.Y.);; 2Division of Cardiology, Department of Internal Medicine, Chang Gung Memorial Hospital-Linkou, Chang Gung University College of Medicine, Taoyuan 33305, Taiwan; 3Department of Mechanical Engineering, Chang Gung University, Taoyuan 33302, Taiwan; 4Department of Chemical and Materials Engineering, Tamkang University, New Taipei City 25137, Taiwan; cthuang@mail.tku.edu.tw

**Keywords:** anti-adhesion, drug-eluting nanofibers, tendon repair, sustained release

## Abstract

The treatment and surgical repair of torn Achilles tendons seldom return the wounded tendon to its original elasticity and stiffness. This study explored the in vitro and in vivo simultaneous release of indomethacin and bupivacaine from electrospun polylactide–polyglycolide composite membranes for their capacity to repair torn Achilles tendons. These membranes were fabricated by mixing polylactide–polyglycolide/indomethacin, polylactide–polyglycolide/collagen, and polylactide–polyglycolide/bupivacaine with 1,1,1,3,3,3-hexafluoro-2-propanol into sandwich-structured composites. Subsequently, the in vitro pharmaceutic release rates over 30 days were determined, and the in vivo release behavior and effectiveness of the loaded drugs were assessed using an animal surgical model. High concentrations of indomethacin and bupivacaine were released for over four weeks. The released pharmaceutics resulted in complete recovery of rat tendons, and the nanofibrous composite membranes exhibited exceptional mechanical strength. Additionally, the anti-adhesion capacity of the developed membrane was confirmed. Using the electrospinning technique developed in this study, we plan on manufacturing degradable composite membranes for tendon healing, which can deliver sustained pharmaceutical release and provide a collagenous habitat.

## 1. Introduction

The Achilles tendon, approximately 15 cm long, is the largest and strongest tendon in humans but is also the most susceptible to rupture [1]. A growing aging population, the increased prevalence of obesity, and the popularization of sports have contributed to an increase in the total incidence of Achilles tendon ruptures, which account for 20% of all large tendon ruptures [2]. High-energy injuries in sports are mainly responsible for rupture among the youth, whereas in the elderly, low-energy injuries, such as the spontaneous rupture of a degenerated Achilles tendon or a rupture in chronic Achilles tendinopathy, are likely causes. An acute tendon injury, either partial or complete, alters the continuity of the tendon and results in loss of movement [1]. Conservative management may be sufficient for young patients with acute sports injuries [3,4]. However, the rupture of a degenerated tendon in older adults demands a different therapeutic approach, as the tendon remains vulnerable to re-rupture after operative treatment [5,6,7]. Tendon injuries cause substantial morbidity, even after appropriate non-surgical or surgical treatments [8,9,10]. Additionally, the postoperative outcomes of the repair of a ruptured tendon remain generally unsatisfactory, despite considerable advances in surgical techniques and rehabilitation methods [11,12].

Tendon adhesion is a common phenomenon that occurs during the healing process of injured tendons [13,14]. It involves the abnormal attachment and fusion of the neighboring tissues, resulting in restricted tendon movement and impaired functionality [15,16]. The formation of adhesions can be attributed to a combination of factors, including inflammation, the formation of scar tissue, and an altered extracellular matrix composition [13,14]. The excessive cross-linking of collagen fibers and disorganized tissue remodeling contribute to the development of adhesions [17]. Strategies to mitigate tendon adhesions encompass both surgical and non-surgical approaches, such as early mobilization, physical therapy, and anti-inflammatory agents [14]. Presently, the literature offers limited insights into the advancement of drug-eluting constructs targeting the prevention of adhesion. Chen et al. advocated the application of poly(L-lactide) fibrous films infused with silver nanoparticles and ibuprofen, showcasing their efficacy in thwarting infections and minimizing adhesion [18]. Kao et al. innovatively used anti-adhesive membranes infused with analgesics, ensuring the sustained release of lidocaine and ketorolac to alleviate postoperative pain [19,20]. An ideal anti-adhesion implant should exhibit biocompatibility and biodegradability, facile adherence to injured surfaces, resilience on exuding surfaces, and the capability to deliver pharmaceutical agents for the mitigation of pain and the facilitation of healing processes.

Tissue engineering presents an alternative strategy to address the unmet clinical demand for the therapy and regeneration of damaged organs [21]. Nanotechnology can aid in optimizing the properties of scaffolds and adjusting their biological functionality to mimic the extracellular matrix (ECM) architecture of natural tissues [22]. Biodegradable polymers have been developed and introduced for soft tissue regeneration. Primarily, nanofibrous forms are adopted for artificial ECMs owing to their unique features, including their high surface-to-volume ratio, tunable porosity, and ease of surface functionalization [23]. Nanofibrous scaffolds also provide possibilities for cell seeding and proliferation, as well as new 3D tissue formation [24]. Among the various biodegradable polymers, poly(lactic-co-glycolic acid) (PLGA) is the most extensively researched owing to its ability to reduce inflammatory effects in the tissue during hydrolytic degradation and its capacity for drug delivery [25,26,27]. The polymer-incorporated drug can be used for therapeutic purposes, such as chemotherapy or antimicrobial activity [28,29,30]. The reduced inflammatory process also diminishes the development of adhesions during the tissue’s healing cascade [31].

The main structural protein in most tissues in the human body, collagen, plays an important role in maintaining the biological and structural integrity of the ECM and provides physical support to tissues. It also exhibits low immunogenicity, a porous structure, permeability, good biocompatibility, and biodegradability while regulating the cells’ morphology, adhesion, migration, and differentiation [32,33]. These extraordinary properties render it a promising biomaterial for tissue-regenerating scaffolds.

Indomethacin, a non-selective cyclooxygenase inhibitor, is a conventional non-steroidal anti-inflammatory drug (NSAID) that effectively reduces inflammation and pain [34]. This conventional NSAID has bidirectional influences on tendon healing: an inhibitory effect in the proliferative phase and a promotive effect in remodeling during tendon healing [35,36,37]. Traditionally, indomethacin has been administered intravenously; however, administration via this route requires repetitive daily injections, which may cause systemic side effects such as impairments in gastric and renal function. Additionally, the bioavailability of the administered drug to the target tissue remains unreliable when administered via the intravenous route [38]. Along with treatment drugs, local anesthetics have been widely used in clinical practice to prevent and alleviate perioperative pain. Bupivacaine is the recommended local anesthetic for caudal, epidural, and spinal anaesthetization and is commonly used to manage acute and chronic pain in clinical practice [39].

We developed resorbable indomethacin/bupivacaine-eluting anti-adhesive PLGA/collagen nanofibers via electrospinning and evaluated their efficacy in treating injured Achilles tendons. Electrospinning is an easy and versatile method for preparing nanofibers made of polymeric materials [40]. We evaluated the physicochemical properties of the biomolecule-loaded nanofibrous membranes after manufacturing and assessed the in vitro and in vivo drug-release behavior. The anti-adhesive and healing capacities and mechanical properties of the repaired Achilles tendon were verified in a rat model, accompanied by histological analyses.

## 2. Results

### 2.1. Physical and Chemical Properties of the Biomolecule-Incorporated Nanofibers

#### 2.1.1. Surface Topography

Figure 1 shows the field emission SEM (10,000× magnification) images of pristine nanofibers and biomolecule-embedded PLGA nanofibers. The diameter and average porosity of the pristine nanofibers were 873.27 ± 331.40 nm and 75.54%, respectively, whereas the diameters of the indomethacin-, collagen-, and bupivacaine-loaded PLGA nanofibers were 247.81 ± 156.89, 215.0 ± 99.09, and 185.85 ± 75.20 nm, respectively, and the corresponding average porosities were 80.69%, 99.10%, and 83.32%, respectively. Adding biomolecules during electrospinning decreased the percentage of polymer in the polymeric solution. This reduction made it easier for the solution to be stretched by an external force, and thus the diameters of the spun nanofibers decreased.

#### 2.1.2. Assessment of Hydrophilicity

Wetting angles were assessed for the pure and biomolecule-loaded PLGA nanofibers. The wetting angles for the pure, indomethacin-embedded, collagen-loaded, and bupivacaine-incorporated PLGA nanofibers were 123.36°, 89.22°, 112.94°, and 93.27°, respectively; thus, incorporating water-soluble drugs increased the hydrophilicity of the nanofibrous mats of PLGA.

#### 2.1.3. Assessment of the Chemical Polarity 

Figure 2A depicts the Fourier transform infrared (FTIR) spectra for the pure PLGA, indomethacin, and indomethacin-loaded PLGA nanofibers. The new peak at 1691–1718 cm^−1^ mainly resulted from the C=O bond of indomethacin. The new vibration peak at 1068 cm^−1^ was attributed to the C–Cl bond of the drug. The characteristic peaks at 2900 cm^−1^ and 1270 cm^−1^, resulting from the O–H and C–O bonds, respectively, were enhanced by the addition of indomethacin [41]. Figure 2B shows the FTIR spectra of the pure and collagen-loaded PLGA nanofibers. The absorption band at 3430 cm^−1^ may be due to the O–H stretching vibration of the incorporated pharmaceuticals. The enhanced peaks at 1272, 1172, and 1090 cm^−1^ mainly resulted from the C–O bond of collagen. The new vibration peak at 1754 cm^−1^ was attributed to the C=O vibration. Furthermore, the vibration peak near 2940–3000 cm^−1^ may have resulted from the N–H bonds of the added collagen [42]. Figure 2C shows the FTIR assay results for pure PLGA, bupivacaine, and bupivacaine-loaded PLGA nanofibers. The peak at 1100 cm^−1^ results from the C–O–C vibration of bupivacaine. Incorporation of the drug enhanced the vibration peak of the C–H bond at 2800–2850 cm^−1^. Additionally, the new peak at 3100–3200 cm^−1^ resulted from the N–H bonds of bupivacaine [43]. Thus, the FTIR spectra confirmed that the biomolecules were satisfactorily embedded into the PLGA nanofibers.

#### 2.1.4. Mechanical Strength of the Fabricated PLGA Nanofibers

The mechanical properties of the electrospun nanofibers were evaluated. The experimental data in Table 1 illustrate that despite an inferior elongation at breaking, the biomolecule-loaded PLGA nanofibers exhibited a higher tensile strength than the pure PLGA nanofibers.

#### 2.1.5. In Vitro Drug Elution from PLGA Nanofibers

The in vitro release profiles of indomethacin and bupivacaine from biomolecule-loaded PLGA nanofibers are shown in Figure 3. The first peak in the indomethacin release curve occurred on Day 1, followed by a gradually diminishing discharge. A burst in the release on Day 1 was noted for bupivacaine, followed by a second peak in the release on Day 8 and a subsequent decrease in discharge. Further, the accumulated elution assessments revealed that 97.2% and 85.6% of the loaded indomethacin and bupivacaine, respectively, were released by Day 30.

#### 2.1.6. In Vivo Drug Release of Nanofibers

Figure 3E shows the in vivo elution profiles of indomethacin and bupivacaine from the implanted nanofibers in rats. The elution of both biomolecules revealed a similar trend, with high-concentration elution being sustainable for at least 28 days.

#### 2.1.7. Bioactivity Examination

The rats were housed within the lab-developed animal behavior cage (ABC) to examine the efficacy of the biomolecule-loaded PLGA nanofibers for one week. Compared with normal rats, the total activity counts were significantly lower in the nanofiber (*p* < 0.05) and control (*p* < 0.01) groups (Figure 4A). Although a statistically significant difference was not found between the activity of rats in the nanofiber and control groups, the rats in the nanofiber group showed more total activity counts during the 7 days of assessment.

### 2.2. Assessment of the Specimens

#### 2.2.1. Assessment of Gross Specimens 

The repaired Achilles tendons were retrieved at 8 weeks after the operation. During the retrieval procedure, the experimental tendons showed less adhesion to the surrounding tissues and exhibited a hypertrophic appearance compared with the tendons in the control group. The diameters of the nanofiber-treated tendons were considerably greater than those of the control (5.4 ± 0.2 vs. 2.0 ± 0.3 mm, *p* = 0.008).

#### 2.2.2. Mechanical Strength

The repaired tendons were retrieved 8 weeks after surgery, and their tensile strength was evaluated (Figure 4B). The average maximal load to failure of the tendons treated with the triple-layer biomolecule-loaded PLGA nanofibers was significantly greater than that of the normal tendons (16.1 ± 6.9 and 4.0 ± 4.7, respectively; *p* < 0.01).

#### 2.2.3. Microscopic Evaluations

The H&E-stained specimens showed consistent alignment of the regenerated collagen (Figure 5A). Additionally, copious amounts of round to ovoid tenocytes were distributed along the collagen, as observed by Masson’s trichrome staining (Figure 5B). Standard immunohistochemical (IHC) staining demonstrated a moderate to strong cytoplasmic expression of the target growth factors (Figure 5C–F). The expression of Type I collagen was significantly more abundant than that of Type III collagen, as calculated from the average optical density (0.38 and 0.04, respectively; *p* < 0.01) (Figure 5G,H).

## 3. Discussion

A torn Achilles tendon can be treated surgically or non-surgically. With either treatment method, an injured tendon undergoes a three-phase regeneration process: inflammation, proliferation/repair, and remodeling [44]. However, despite the completion of the remodeling process, the healed tendon often has scar-like tissue and never completely regains its biomechanical properties [44,45]. The depletion of tenocytes and the replacement of Type I collagen by Type III may be the major cause of this change [46,47,48]. Several advanced treatments targeting the repair of Achilles tendon injuries other than suture-alone repair have been described [49,50,51,52,53,54]. Additionally, researchers have advocated restoration of the normal biological and physical properties of the tendon by upregulating cellular and tissue responses during tendon repair, such as supplementing bioactive growth factors, modulating the inflammatory response, and tissue engineering [55,56,57,58,59].

We proposed a novel scheme using triple-layered PLGA nanofibers incorporated with three biomolecules (indomethacin, collagen, and bupivacaine) for the treatment and regeneration of injured Achilles tendons. In the presence of indomethacin and collagen, the repaired tendon tissues primarily comprised Type I collagen, with an abundance of tenocytes. The repaired tendons also exhibited extraordinary mechanical performance. Moreover, the activity of the rats rapidly returned to normal following the administration of bupivacaine.

Indomethacin is a conventional NSAID that inhibits cyclooxygenase (COX)-1 and COX-2, thereby decreasing tissue inflammation and reducing pain. Although an animal study reported that indomethacin negatively affected the tensile strength and stiffness of tendons after healing [35], other reports indicated its positive contribution [36,37,60,61]. Forslund et al. injected rats with indomethacin at different doses (1.5, 3.9, and 5.0 mg/kg) daily and found that the tendons regenerated well; notably, the tensile strength of the tendons increased considerably in one experimental group [36]. Moreover, an earlier animal study revealed no significant differences in the tensile force to failure or histological appearance under daily administration of indomethacin (1.5 mg/kg) [37]. The microscopic evidence presented by Mallick et al. suggested that at a therapeutic dose, indomethacin did not inhibit the proliferation of tenocytes [61]. In addition to the increased number of tenocytes, growth factors, such as vWF, VEGF, TGF-β, and BMP-2, were abundant under the stimulation of sustained indomethacin elution. Moreover, the amount of Type I collagen was approximately 10 times that of Type III, demonstrating that indomethacin has positive effects on tendon tissue’s regeneration. As the biomolecule-loaded nanofibers provided a sustained discharge of indomethacin for over 30 d, there was no need for repeated administration (per os or intravenous) of indomethacin.

In addition to the supplemental drugs that stimulate the regeneration of an injured tendon, the scaffold for the recruitment of tenocytes during tendon regeneration is also crucial. The mechanism of tendon healing is a complex process that requires the expression of various growth factors and cytokines, including the existence of an ECM to facilitate angiogenesis and the arrangement and proliferation of tenocytes [62,63,64]. When contrasted with other drug-eluting films, such as patterned microcontainer films [65], electrospun nanofibers exhibit a structure that closely resembles the ECM, thereby promoting tissue healing. Biodegradable nanofibers incorporated with collagen as a scaffold can stimulate the differentiation of mesenchymal stem cells into functional cells, such as osteoblasts and chondrocytes [66,67,68,69]. The incorporation of water-soluble drugs increased the hydrophilicity of the nanofibrous mats of PLGA, which, in turn, further enhanced tissue healing. The empirical data obtained here indicated that the recruited cells and collagens were well regulated. The recruited cells were theoretically transformed into tenocytes under a high concentration of the resulting growth factors. Thus, the superior mechanical strength of the tested tendons was realized by its rich composition of tenocytes and Type I collagen.

Currently, supplemental materials that contain anti-adhesive properties after tendon repair/reconstruction procedures have gained popularity [31,70,71]. During the early phase of tendon healing, an inflammatory process accounts for recruiting inflammatory cells and other cytokines [44]. This process does not help the repair process of the wounded tendon; on the contrary, it causes adhesion of the surrounding tissues. Meanwhile, the protective immobilization of the operated joint may also have an effect on the postsurgical adhesion [72]. The supplement of the nanofibrous PLGA membrane may provide a shield that prevents the skin from directly contacting the repaired tendon during the immobilization period. PLGA has found extensive use in drug delivery applications. However, a notable drawback of PLGA is its relatively low mechanical strength. Alternative degradable polymers such as polycaprolactone [73] can also be considered for strengthening tendons during the healing process. Meanwhile, the addition of indomethacin also reduces the inflammatory process and thus may reduce the adhesive process during tendon healing. Among the various biodegradable polymers, PLGA is the most extensively researched, owing to its ability to reduce inflammatory effects in the tissue during hydrolytic degradation and its capacity for drug delivery [25,26,27].

Another significant contribution of this study was the pain-controlling effect provided by the sustained release of bupivacaine from the PLGA nanofibers. Postoperative pain, particularly in orthopedic surgeries, is a major concern. Although analgesic agents are systemically or locally administered routinely after surgery, they have drawbacks such as inadequate bioavailability or adverse systemic effects [74,75,76]. In the current study, an analgesic biomolecule (bupivacaine) was incorporated into the nanofibrous mats and co-eluted with the other biomolecules. The rats in the nanofiber group regained their preoperative activities faster than those in the control group. The in vitro and in vivo results showed that the elution of the embedded bupivacaine was sustained for at least 28 days. Numerous studies have investigated the incorporation of various cell types into biodegradable scaffolds to enhance the healing of Achilles tendons [77,78]. However, by embedding a cost-effective NSAID, i.e., indomethacin, with bupivacaine into a rapidly degrading scaffold, we obtained favorable outcomes. This approach not only promoted the healing of the tendon and the restoration of its flexibility and strength, but also effectively provided pain relief to facilitate the recovery of physical activity.

## 4. Materials and Methods

### 4.1. Materials and Fabrication of Built-in-Biomolecule PLGA Nanofibers

The polymeric material was a commercially available PLGA polymer comprising lactide/glycolide (50/50) with a molecular weight of 24,000–33,000 Da (Resomer RG 503; Boehringer Ingelheim, Ingelheim, Germany). The biomolecules—indomethacin, collagen, and bupivacaine—were acquired from Sigma-Aldrich (Saint Louis, MO, USA). To manufacture indomethacin- and bupivacaine-incorporated PLGA/collagen nanofibers, PLGA/indomethacin (1120 mg/280 mg), PLGA/collagen (1120 mg/0.2 mL), and PLGA/bupivacaine (1120 mg/280 mg) were mixed with the solvent 1,1,1,3,3,3-hexafluoro-2-propanol (HFIP; 5 mL) using a magnetic stirrer for 4 h. The solutions were electrospun and deposited layer by layer onto the collector (an aluminum plate measuring 15 cm × 15 cm) using a lab-made device consisting of a syringe/needle (internal diameter: 0.42 mm), a grounded collector, and a high-voltage power supply (DC voltage, 36 kV; current, 4.16 mA). A controlled environment (ambient temperature, 27 °C; humidity, 65–68%) was maintained throughout the manufacturing process. The voltage, the delivery rate of the syringe, and the distance from the syringe to the collector were 17 kV, 0.6 mL/h, and 15 cm, respectively. A triple-layered nanofibrous membrane comprising PLGA–indomethacin, PLGA–collagen, and PLGA–bupivacaine layers was fabricated. To evaporate the solvents, all the nanofibrous mats generated from PLGA were kept in a chamber at 40 °C for 72 h.

### 4.2. Physicochemical Properties of the Triple-Layered PLGA Nanofibers

#### 4.2.1. Surface Topography

The structure of the spun triple-layered nanofibrous mats of PLGA was evaluated using field emission scanning electron microscopy (SEM, JSM–7500F, Jeol, Tokyo, Japan) after coating with gold. The fiber size, size distribution, and porosity were assessed using SEM images generated via ImageJ software Version 1.53t (National Institutes of Health, Bethesda, MD, USA) [79,80]. The densities of the electrospun nanofibers were calculated by dividing their mass by their volume. The apparent porosity of the nanofibrous membranes was calculated using the following equation
(1)$$\mathrm{Pore}(\%)=\left\{1-\frac{\rho_{\mathrm{membrane}}}{\rho_{\mathrm{polymer}}}\right\}$$
where ρ_membrane_ and ρ_polymer_ denote the densities of the nanofibrous membrane and the polymer, respectively.

#### 4.2.2. Assessment of Hydrophilicity 

The hydrophilicity of the spun triple-layered nanofibrous mats of PLGA was evaluated using a water contact angle analyzer (First Ten Angstroms, Portsmouth, VA, USA). A 10 mm × 10 mm polymer specimen was placed on the testing plate, and distilled water was slowly dripped onto its surface. The contact angles between the droplets and the nanofibrous samples were assessed using a video check.

#### 4.2.3. Assessment of Chemical Polarity 

The manufactured polymers were evaluated using FTIR spectrometry. A Nicolet iS5 spectrometer (Thermo Fisher Scientific, Waltham, MA, USA) was operated at a resolution of 4 cm^−1^, and 32 scans were used to complete the FTIR assay. Nanofibrous samples were compressed into KBr disks and assayed in the 400 to 4000 cm^−1^ range.

#### 4.2.4. Mechanical Strength of the Fabricated PLGA Nanofibers

All the fabricated PLGA nanofibers were characterized using a Lloyd tester (Ametek, Berwyn, PA, USA) with a 2.5 kN load cell to determine their tensile strength. Rectangular samples (20 mm × 50 mm) were cut from the electrospun nanofibers for the experiments. The variations in stress–strain were recorded by applying a constant extension rate of 60 mm·min^–1^ to the samples.

#### 4.2.5. In Vitro Elution Characterization of Drug-Incorporated Nanofibers

An in vitro elution scheme was used to determine the discharge characteristics of indomethacin and bupivacaine from the PLGA mats. Before quantification, the nanofibrous samples were submerged in a dissolution medium comprising a phosphate buffer (0.15 mol·L^–1^, pH 7.4). First, samples with a controlled area (20 mm × 30 mm) were cut and incubated in a test tube containing 1 mL of phosphate-buffered saline at a constant temperature (37 °C) for 24 h. Subsequently, the tested medium was isolated, collected, and analyzed at 24 h intervals. The phosphate buffer (1 mL) was replaced every 24 h until the sample was completely dissolved. The pharmaceutical concentration in the buffer was determined by high-performance liquid chromatography (HPLC) using a Hitachi L-2200 Multisolvent Delivery System (Tokyo, Japan). A Mighty RP-18GP (150 mm × 4.6 mm, 5 μm) column was used for the assay. The mobile phase for indomethacin was acetonitrile and distilled water (pH 3) at a ratio of 70/30 (*v*/*v*). The absorbency was monitored at 260 nm, with a flow rate of 1.25 mL min^–1^. The retention time was 2.5 min. Meanwhile, the mobile phase used for bupivacaine contained acetonitrile, distilled water, and orthophosphoric acid (70:30:0.1) (*v*/*v*/*v*). The absorbency was 210 nm, and the flow rate was 1 mL min^–1^. The retention time was 2.5 min.

#### 4.2.6. Animal Model Simulating Achilles Tendon Injury/Repair

Twenty-one male Sprague–Dawley rats weighing 250 ± 20 g were used for the animal study. All animal experiments were approved by the Institutional Animal Care and Use Committee, and the animals were handled in accordance with the guidelines and regulations of the Ministry of Health and Welfare of Taiwan. The animals were anesthetized in a transparent acrylic box (40 cm × 20 cm × 28 cm) with isoflurane (Aesica-Queenborough, Queenborough, Kent, UK) via a vaporizer (Midmark, Versailles, OH, USA). Inhalational anesthesia was maintained throughout the operational process. For analgesia and hemostasis, the rats received a local injection over the right leg containing 0.5 mL of 2% Xylestesin-A with epinephrine at a concentration of 1:100,000 before the surgical procedure. The right leg was shaved and prepared using the standard antiseptic procedure. A 3 cm straight cut of the skin parallel to the Achilles tendon was created. The Achilles tendon was exposed by dissecting the subcutaneous fat and surrounding soft tissue. The mid-portion of the Achilles tendon was transected and then sealed end-to-end using a 5-0 Vicryl suture. Subsequently, the animals were arbitrarily divided into three groups: normal (*n* = 3), control (*n* = 9), and nanofiber (*n* = 9).

No surgery was performed on the rats in the normal group, rats in the control group underwent surgery only (with no additional nanofibers), and rats in the nanofiber group underwent surgery followed by wrapping triple-layered nanofibers (inner layer, indomethacin; middle layer, collagen; outer layer, bupivacaine) circumferentially around the injured Achilles tendons of the study rats (Figure 6). After implantation, the wound was sealed with a 3-0 nylon suture. A topical bactericidal ointment was applied to the surgical wound to prevent infection. The rats were returned to their cages after the effects of anesthesia wore off.

#### 4.2.7. Assessment of Bioactivity

The postsurgical activity level and the food and water consumption of the rats were monitored daily by keeping each animal in a lab-developed ABC (50 cm × 50 cm × 50 cm) for 7 d. The cage was symmetrically separated into 9 areas, each equipped with a photoelectric switch sensor (HP100-A1, Azbil Corp Tokyo, Tokyo, Japan) on the top to monitor and record movements; the sensors were placed at a distance of 16.7 cm from each other. The sensor was triggered by the arrival of the rat and started recording as the rat moved. The total number of times the sensors were triggered was monitored using a computer equipped with an acquisition interface. A constant temperature (22–25 °C), pressure (1 atm), and humidity (60–70%) were maintained during the entire study period. The animals were returned to their initial cage after the bioactivity assessment.

#### 4.2.8. Characterization of Drug Elution In Vivo 

To evaluate the in vivo release profiles of the triple-layered PLGA nanofibers, tissue samples adjacent to the target Achilles tendon were obtained from each rat in the study group at weekly intervals until the rats were euthanized. The anaesthetization and surgical procedures were similar to those for the implantation of the nanofibrous membrane. The collected specimens were preserved in a 10% formalin solution and analyzed using HPLC. All the obtained specimens were analyzed without exclusions.

### 4.3. Evaluation of the Specimens 

#### 4.3.1. Evaluation of the Gross Specimens 

The Achilles tendons of the specimens were excised after euthanasia for further experiments. The cross-sectional diameters of the tendons where the previous transections had been performed were measured using digital calipers.

#### 4.3.2. Evaluation of the Mechanical Strength 

The mechanical strength of the Achilles tendons was measured using a Lloyd tensiometer (AMETEK, Berwyn, PA, USA) with a 2.5 kN load cell. The tendons were initially wrapped with gauze immersed in a normal saline solution to simulate the in vivo environment. During measurement, the repaired Achilles tendons were extended at a rate of 60 mm·min^–1^, and the resulting load–extension curves were recorded (*n* = 3). The average maximum strength was calculated and used for the subsequent analysis.

#### 4.3.3. Microscopic Evaluations

The collected tendons were preserved in 10% phosphate-buffered formalin and sliced into fragments 2 mm wide. They were processed and embedded in paraffin. Tissue sections with a thickness of 4 μm were acquired for histological assessments using a microtome (Sakura Finetek, Tokyo, Japan). The specimens were blotted with hematoxylin and eosin (H&E) and Masson’s trichrome stains. The arrangement of collagen and the activity of the tenocytes were observed under a microscope with magnification of up to 400×.

Standard IHC staining was performed on 4 μm paraffin sections of the tendon tissue to verify the expression of growth factors, such as bone morphogenetic protein (BMP2), vascular endothelial growth factor (VEGF), von Willebrand factor (vWF), transforming growth factor beta (TGF-β), and Type I and III collagens. Commercial antibodies were used for the IHC analyses: BMP2 (polyclonal, 1:50, A0231, ABclonal, Houston, TX, USA), VEGF (polyclonal, 1:100, A0280, ABclonal), vWF (vWF Picoband™ antibody, 1:200, PB9062, Boster Biological Technology, Pleasanton, CA, USA), TGF-β (polyclonal, 1:50, A2561, ABclonal) Type I collagen (1:2000, A1352, ABclonal), and Type III collagen (1:400, A00788-3, ABclonal). Thermally induced epitope retrieval was performed (95 °C/30 min) in a citrate buffer (pH = 6.0) after deparaffinization and rehydration. The endogenous peroxidase was quenched in a 3% hydrogen peroxide solution for 5 min, followed by rinsing in phosphate-buffered saline. Subsequently, the tissue sections were repaired for 40 min using ethylenediaminetetraacetic acid. The slides were incubated with the primary antibody for 60 min at 37 °C and overnight at 4 °C thereafter. After rinsing thrice in the buffer, the slides were incubated with the secondary antibodies. Tissues were stained using a 3,3′-diaminobenzidine substrate chromogen solution. To quantify the Type I and III collagens in the specimens, the average optical density was processed using ImageJ software (National Institutes of Health, Bethesda, MD, USA).

### 4.4. Statistical Analyses

Statistical analyses were performed using SPSS software (V26.0 Windows, SPSS Inc., Chicago, IL, USA). The acquired data are presented as the mean ± standard deviation and compared via the paired *t*-test. The statistical significance was set at a *p*-value of <0.05.

## 5. Conclusions

In conclusion, we successfully developed triple-layered biomolecule-loaded PLGA nanofibers via electrospinning that could be implanted during surgical repair to treat and regenerate ruptured Achilles tendons. The prepared nanofibers provided a sustained discharge of indomethacin and bupivacaine for at least 28 days in vivo and facilitated the healing of the injured Achilles tendon. The tendons also regenerated well, resulting in superior mechanical strength and rapid recovery of activity. The novel PLGA–collagen scaffold can potentially improve tendon healing because of its capability to deliver pharmaceuticals while simultaneously providing a collagenous habitat in an operationally convenient treatment. Despite the favorable outcomes of our study, certain limitations existed. The first limitation pertains to the relatively small number of animals enrolled. Second, dose-dependent experiments with the built-in drugs were not performed in vitro. Further studies should elucidate the elution profiles at different doses. Additional work is also required to optimize both the distribution of the fibers’ sizes and the associated drug-releasing behavior. Third, the tenocytes were only confirmed by a microscopic examination using Masson’s trichrome staining. Cell culture and mRNA certification were not conducted in this study. Fourth, we did not systemically examine the concentration of the loaded drug. The systemic side effects of this therapy, such as cardiac or renal involvement, require evaluation. Finally, although we tried to mimic a wounded Achilles tendon in humans with harp scalpel-cut transections in rats, a torn Achilles tendon after an unexpected distracted axial force results in a degenerated tendon, in which the healing process would be slower or inadequate. A degenerative Achilles tendon animal model is needed to test this hypothesis further.

## Figures and Tables

**Figure 1 ijms-24-16235-f001:**
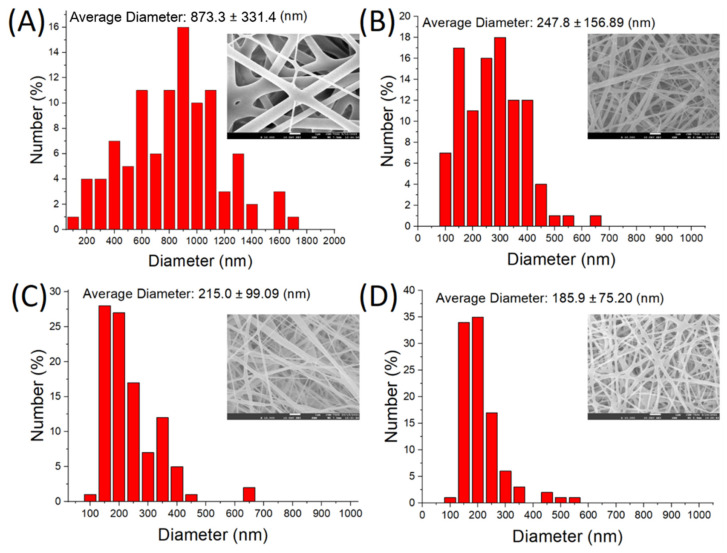
Field emission SEM (10,000× magnification) images and size distributions of the biomolecule-embedded PLGA nanofibers: (**A**) pure, (**B**) indomethacin-loaded, (**C**) collagen-embedded, and (**D**) bupivacaine-incorporated PLGA.

**Figure 2 ijms-24-16235-f002:**
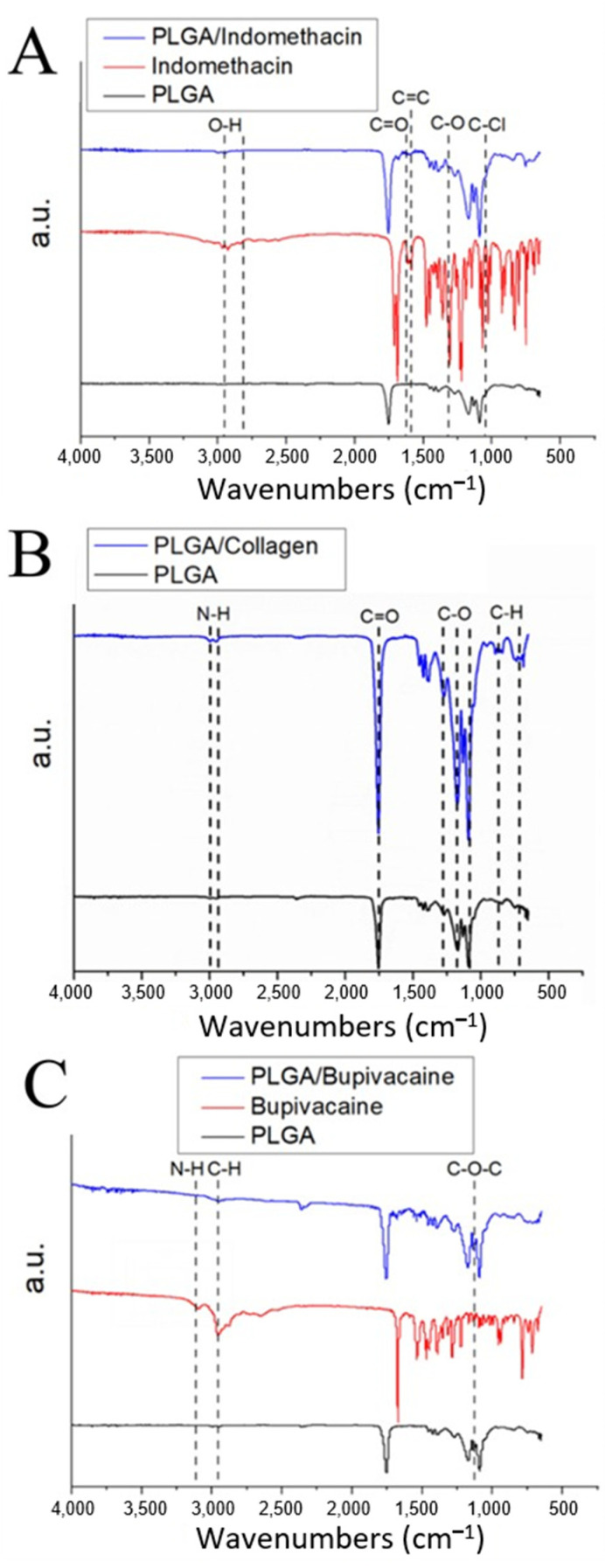
FTIR spectra for (**A**) pure PLGA, indomethacin, and indomethacin-loaded PLGA nanofibers; (**B**) pure and collagen-loaded PLGA nanofibers; and (**C**) pure PLGA, bupivacaine, and bupivacaine-loaded PLGA nanofibers.

**Figure 3 ijms-24-16235-f003:**
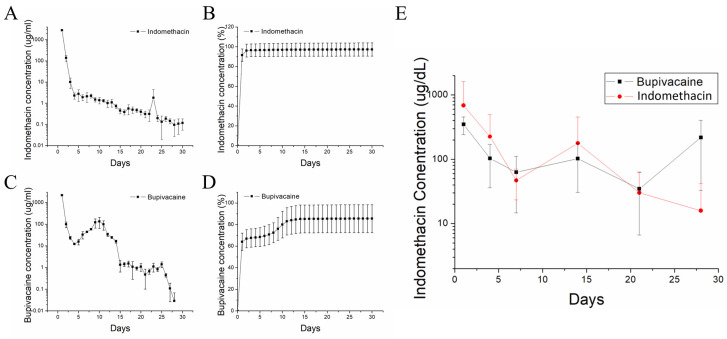
In vitro release profiles of indomethacin ((**A**) daily and (**B**) cumulative) and bupivacaine ((**C**) daily and (**D**) cumulative) from biomolecule-loaded PLGA nanofibers. (**E**) In vivo elution profiles of indomethacin and bupivacaine from the implanted nanofibers in rats.

**Figure 4 ijms-24-16235-f004:**
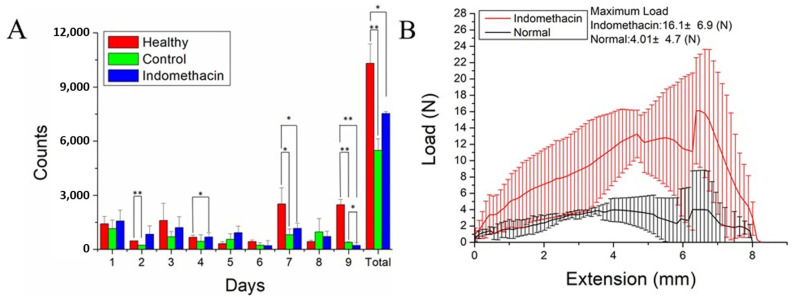
(**A**) Total activity counts of the study rats within 1 week of the treatment. No statistically significant differences between the indomethacin (blue bar) and control (green bar) groups were found, but the indomethacin group exhibited higher overall activity throughout the 7-day assessment. (**B**) Tensile strength of repaired tendons that were retrieved 8 weeks after surgery. * *p* < 0.05; ** *p* < 0.01.

**Figure 5 ijms-24-16235-f005:**
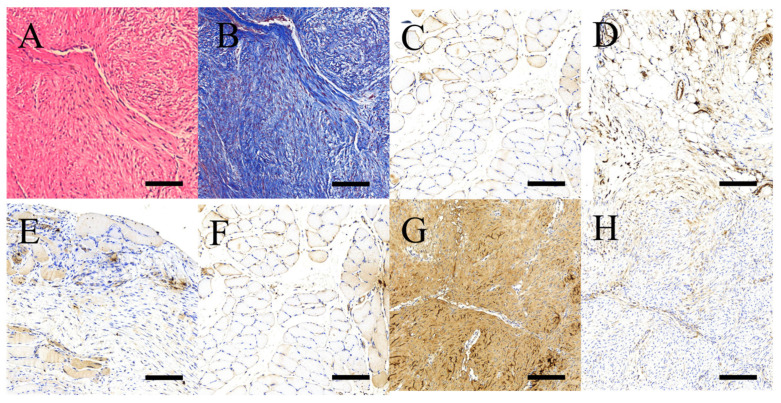
Representative micrographs of histological tendon sections. (**A**) H&E staining, (**B**) Masson’s trichrome staining, (**C**) BMP-2, (**D**) VEGF, (**E**) vWF, (**F**) TGF-β, (**G**) Type I collagen, and (**H**) Type III collagen. Scale bar, 500 μm.

**Figure 6 ijms-24-16235-f006:**
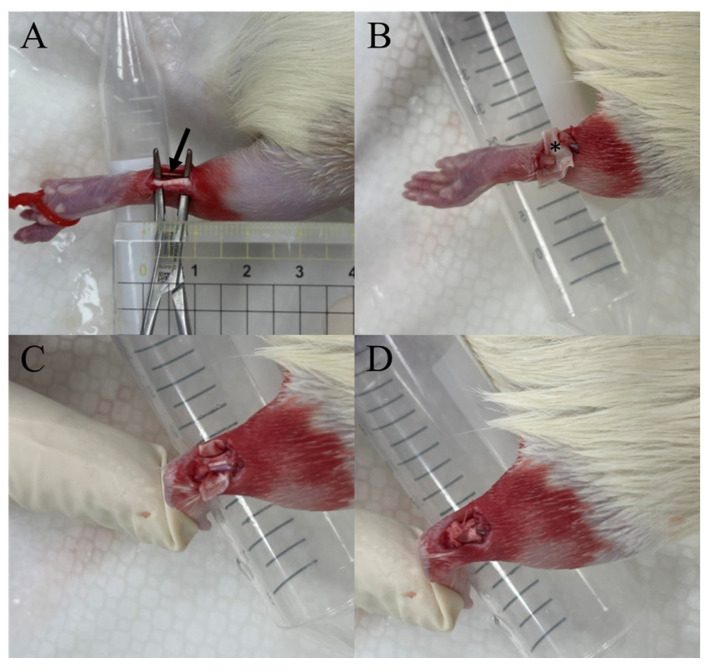
Wrapping of triple-layered nanofibers (inner layer, indomethacin; middle layer, collagen; outer layer, bupivacaine) circumferentially around the injured Achilles tendons of the study rats. (**A**) Locating the Achilles tendon (indicated by the arrow) and making an incision at its midpoint. (**B**) Positioning the triple-layered nanofibers (*) beneath the cut tendon. (**C**) Stitching the ends of the tendon together. (**D**) Enveloping the nanofibers around the tendon in a circular manner.

**Table 1 ijms-24-16235-t001:** Mechanical properties of the electrospun nanofibers.

Type of Nanofibers	Maximum Strength (MPa)	Elongation at Breaking (%)
PLGA	1.13 ± 0.1	160.85 ± 7.82
PLGA/collagen	1.06 ± 1.01	30.32 ± 0.21
PLGA/bupivacaine	2.69 ± 0.19	83.64 ± 1.95
PLGA/indomethacin	2.03 ± 1.42	74.43 ± 7.86

PLGA: poly (lactic-co-glycolic acid).

## Data Availability

All data generated or analyzed during this study are included in this published article. The datasets used and/or analyzed during the current study are available from the corresponding author upon reasonable request.

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
