# Peer review of "Anti-Adhesive Resorbable Indomethacin/Bupivacaine-Eluting Nanofibers for Tendon Rupture Repair: In Vitro and In Vivo Studies"

_ijms, 2023, doi:10.3390/ijms242216235_

Round 1

Reviewer 1 Report

Comments and Suggestions for Authors

The authors present a novel way to repair tendons via a drug releasing polylactic fiber system. The work is novel and important for medical applications and paves the way for similar systems. The following points should be addressed:

1.       Page 1 line 37 reference missing.

2.       Page 1 line 41 reference missing.

3.       Page 1 line 42 reference missing.

4.       Page 1 line 44 reference missing.

5.       Page 2 line 47 reference missing.

6.       Page 2 line 49 reference missing.

7.       Page 2 line 50 reference missing.

8.       Page 2 line 69 reference missing. I suggest 1

9.       Page 2 line 72 reference missing. I suggest 2, this system is not only biocompatible, it is even antibacterial, allows cell seeding and proliferation.

10.   Page 2 line 86 reference missing.

11.   Page 2 line 94 reference missing.

12.   Figure 1: The porosity of the scaffolds need to be calculated and presented.

13.   The graphs in Figure 2 are relatively small compared to the labels, maybe the graphs could be increased for visibility…

14.   ImagesJ requires to cite its papers as the agreement for usage, the citations are: 3,4

15.   The authors mention drug releasing films covered with collagen. There is also a 2.5D system offering such a possibility.5 The authors should state such a competing system for this reason, especially since already companies like the Swiss Incaptek sell it https://incaptek.com.

16.   Eluent of the HPLC as well as column and flow rate need to be stated. In addition, the detection mechanism of the HPLC is missing, and company HQ, country.

17.   Figure 5: Scale bars are missing.

18.   Figure 6 could be more informative by adding of what is to seen here.

19.   What about the wettability and roughness of all sample types. These parameters should also be investigated and connected with the biological results.

20.   For the HPLC: What solvents were used in which profile and for which time for the drug detection?

21.   Why was PLGA used in this study? It is known that the mechanical properties of PLGA scaffolds are not very good. Which alternative materials could be used for this purpose?

22.   How was the residual solvent HFIP removed from the scaffolds? HFIP is toxic, which why it has to be removed.

23.   For the electrospinning process: the spinneret speed and collector rotation speed, as well as the collector dimensions are missing.

References

(1)         Lou, C.-W.; Yao, C.-H.; Chen, Y.-S.; Hsieh, T.-C.; Lin, J.-H.; Hsing, W.-H. Manufacturing and Properties of PLA Absorbable Surgical Suture. Text. Res. J. 2008, 78 (11), 958–965. https://doi.org/10.1177/0040517507087856.

(2)         Badaraev, A. D.; Lerner, M. I.; Bakina, O. V.; Sidelev, D. V.; Tran, T.-H.; Krinitcyn, M. G.; Malashicheva, A. B.; Cherempey, E. G.; Slepchenko, G. B.; Kozelskaya, A. I.; Rutkowski, S.; Tverdokhlebov, S. I. Antibacterial Activity and Cytocompatibility of Electrospun PLGA Scaffolds Surface-Modified by Pulsed DC Magnetron Co-Sputtering of Copper and Titanium. Pharmaceutics 2023, 15 (3), 939. https://doi.org/10.3390/pharmaceutics15030939.

(3)         Abràmoff, M. D.; Magalhães, P. J.; Ram, S. J. Image Processing with ImageJ Part II. Biophotonics Int. 2005, 11 (7), 36–43.

(4)         Abràmoff, M. D.; Magalhães, P. J.; Ram, S. J. Image Processing with ImageJ. Biophotonics Int. 2004, 11 (7), 36–42.

(5)         Gai, M.; Kurochkin, M. A.; Li, D.; Khlebtsov, B.; Dong, L.; Tarakina, N.; Poston, R.; Gould, D. J.; Frueh, J.; Sukhorukov, G. B. In-Situ NIR-Laser Mediated Bioactive Substance Delivery to Single Cell for EGFP Expression Based on Biocompatible Microchamber-Arrays. J. Control. Release 2018, 276 (28), 84–92. https://doi.org/10.1016/j.jconrel.2018.02.044.

Author Response

Reviewer #1

The authors present a novel way to repair tendons via a drug releasing polylactic fiber system. The work is novel and important for medical applications and paves the way for similar systems. The following points should be addressed:

  1. Page 1 line 37 reference missing.

Response: Thank you for the comment. Reference has been added.

  1. Page 1 line 41 reference missing.

Response: Thank you for the comment. Reference has been added.

  1. Page 1 line 42 reference missing.

Response: Thank you for the comment. Reference has been added.

  1. Page 1 line 44 reference missing.

Response: Thank you for the comment. Reference has been added.

  1. Page 2 line 47 reference missing.

Response: Thank you for the comment. Reference has been added.

  1. Page 2 line 49 reference missing.

Response: Thank you for the comment. Reference has been added.

  1. Page 2 line 50 reference missing.

Response: Thank you for the comment. Reference has been added.

  1. Page 2 line 69 reference missing. I suggest 1

Response: Thank you for the comment. Reference has been added.

  1. Page 2 line 72 reference missing. I suggest 2, this system is not only biocompatible, it is even antibacterial, allows cell seeding and proliferation.

Response: Thank you for the comment. Reference has been added.

  1. Page 2 line 86 reference missing.

Response: Thank you for the comment. Reference has been added.

  1. Page 2 line 94 reference missing.

Response: Thank you for the comment. Reference has been added.

  1. Figure 1: The porosity of the scaffolds need to be calculated and presented.

Response: Thank you for the comment. The porosities of the scaffolds have been calculated and presented (lines 111-115, lines 320-327).

  1. The graphs in Figure 2 are relatively small compared to the labels, maybe the graphs could be increased for visibility…

Response: Thank you for the comment. Figure 2 has been enlarged for better visibility.

14.ImagesJ requires to cite its papers as the agreement for usage, the citations are:3,4

Response: Thank you for the comment. The two references have been cited (line 320).

  1. The authors mention drug releasing films covered with collagen. There is also a 2.5D system offering such a possibility.5The authors should state such a competing system for this reason, especially since already companies like the Swiss Incaptek sell it https://incaptek.com.

Response: Thank you for the comment. When contrasted with other drug-eluting films, like patterned microcontainer films [65], electrospun nanofibers exhibit a structure that closely resembles the ECM, thereby pro-moting tissue healing. The text has been amended to enhance the quality of the discussions. (lines 253-256).

  1. Eluent of the HPLC as well as column and flow rate need to be stated. In addition, the detection mechanism of the HPLC is missing, and company HQ, country.

Response: Thank you for the comment. The pharmaceutical concentration in the buffer was determined by high-performance liq-uid chromatography (HPLC) using a Hitachi L-2200 Multisolvent Delivery System (Tokyo, Japan). A Mighty RP-18GP (150 mm*4.6 mm, 5 mm) column was used for the assay. The mobile phase for indomethacin was acetonitrile:distilled water (pH:3) in a ratio of 70/30 (v/v). The absorbency was monitored at 260 nm, with a flow rate of 1.25 mL min–1. The re-tention time was at 2.5 mins. Meanwhile, the mobile phase used for bupivacaine con-tained acetonitrile:distilled water:orthophosphoric acid (70:30:0.1) (v/v/v). The absorbency was 210 nm and the flow rate was 1 mL min–1. The retention time was at 2.5 mins. The manscript has been revised to provide these information (lines 355-361).

  1. Figure 5: Scale bars are missing.

Response: Thank you for the comment. Scale bar has been added in Figure 5.

  1. Figure 6 could be more informative by adding of what is to seen here.

Response: Thank you for the comment. The caption of Figure 6 has been revised to better address the images (lines 388-391).

  1. What about the wettability and roughness of all sample types. These parameters should also be investigated and connected with the biological results.

Response: Thank you for the comment. The wetting angles for the pure, indomethacin-embedded, collagen-loaded, and bupiva-caine-incorporated PLGA nanofibers were 123.36°, 89.22°, 112.94°, and 93.27°, respective-ly; thus, incorporating water-soluble drugs increased the hydrophilicity of the PLGA nan-ofibrous mats. The incorporation of water-soluble drugs increased the hydrophilicity of the PLGA nano-fibrous mats, which in turn further enhances tissue healing. The manuscript has been revised to better address the experimental results (lines 125-128, lines 258-259).

  1. For the HPLC: What solvents were used in which profile and for which time for the drug detection?

Response: Thank you for the comment. A Mighty RP-18GP (150 mm*4.6 mm, 5 mm) column was used for the assay. The mobile phase for indomethacin was acetonitrile:distilled water (pH:3) in a ratio of 70/30 (v/v). The absorbency was monitored at 260 nm, with a flow rate of 1.25 mL min–1. The re-tention time was at 2.5 mins. Meanwhile, the mobile phase used for bupivacaine con-tained acetonitrile:distilled water:orthophosphoric acid (70:30:0.1) (v/v/v). The absorbency was 210 nm and the flow rate was 1 mL min–1. The retention time was at 2.5 mins. The manscript has been revised to provide these information (lines 355-361).

  1. Why was PLGA used in this study? It is known that the mechanical properties of PLGA scaffolds are not very good. Which alternative materials could be used for this purpose?

Response: Thank you for the comment. PLGA has found extensive use in drug delivery applications. However, a notable draw-back of PLGA is its relatively low mechanical strength. Alternative degradable polymers like polycaprolactone [73] can also be considered for strengthening tendons during the healing process. The text has been revised to enhance the quality of the discussions (lines 272-275).

  1. How was the residual solvent HFIP removed from the scaffolds? HFIP is toxic, which why it has to be removed.

Response: Thank you for the comment. To evaporate the solvents, all the generated PLGA nanofibrous mats were kept in a cham-ber at 40 oC for 72 h (lines 312-313).

  1. For the electrospinning process: the spinneret speed and collector rotation speed, as well as the collector dimensions are missing.

Response: Thank you for the comment. The collector (an aluminum plate of 15 cm x 15 cm) is a stationary one. The voltage, delivery rate of the syringe and the distance from the syringe to the collector were 17 kV, 0.6 mL/h, and 15 cm, respectively (lines 305, 309-310).

Reviewer #2

In this paper, the authors have developed a new type of nanofibrous composite material through the use of the electrospinning method that heals ruptured tendons. The article is overall written well, and the conclusions are well based on the experimental data. Some improvements to the text should be made, though:

1/ The materials and methods section should be placed before the Results and the Discussion.

Response: Thank you for the comment. Following the format of the Journal, the Materials and Methods section was placed after the the Results and the Discussion.

2/ Figure 2 should be presented more clearly.

Response: Thank you for the comment. Figure 2 has been enlarged for better visibility.

3/ There are some inconsistencies in the Section 2.2.2. that the authors should address. The maximal load for the normal tendons looks wrong, as the error seems larger than the average value, and the values in the text don't match the values in Figure 4B. In addition, Figure 4 should be better explained in either the text or the caption, and the 4.3.2 section doesn't match this figure in terms of solely analyzing the averaged curve.

Response: Thank you for the comment. Indeed, significant errors stemmed from the nature of animal tests. Consequently, average values were employed for comparing different groups. To enhance clarity, we have modified "maximal load" to read as "average maximal load" (line 190). Furthermore, we've updated the caption of Figure 4 to more accurately describe the experimental data.

4/ Further, the authors should compare the results of their study with other similar studies in the field.

Response: Thank you for the comment. Numerous studies have investigated the incorporation of various cell types into biodegradable scaffolds to enhance Achilles tendon healing [77-79]. However, by embedding a cost-effective NSAID—indomethacin, with bupivacaine into a rapidly degrading scaffold, we obtained favorable outcomes. This approach not only promoted tendon healing and the restoration of its flexibility and strength but also effectively provided pain relief to facilitate the recovery of physical activity. The text has been revised to enhance the discussions (lines 288-293).

Reviewer #3

As showed in Figure 1, the diameter present high values of standard deviation, specifically for indomethacin loaded nanofibers. How can the authors justify this phenomenom? This is a crucial point which could negatively affect the release kinetics of the drug, as there are nanofibers with many different diameters.

Response: Thank you for the comment. Various parameters (PLGA to drug ratio, solvent type, voltage, etc.) may affect the distribution of fiber size. Additional works are required to optimize both the distribution of fiber sizes and the associated drug release behavior. This will be the topic of our future works. We have modified the manuscript to include this limitation (lines 462-464).

In the conclusions, the authors state that the prepared nanofibers provided a sustained discharge of indomethacin for at least 30 days. However, according to the results, as showed in Figure 3 (A and B), the authors affirmed that the first peak in the indomethacin release curve occurred on day 1, which corresponds almost to 100%. So, it would not be too appropriate consider the nanofiber as sustained drug delivery for indomethacin for the time frame reported in this study (30 days). Indeed, in only five days, the concentration is decreased by 1000 times. Could this condition be beneficial in the view of a translation of clinical administration? Could please the author provide literature regarding the administrated dose of the drug?

Response: Thank you for the comment. In vitro, the drugs were released fastly in the first few days. Luckily, the in vivo curve showed a sustained and more uniform release of the pharmaceuticals (Figure 3E), mainly due to the fact that the metobolic rate in vivo is slower. This provides advantage in terms of a translation of clinical administration. Additional works are also required to optimize both the distribution of fiber sizes and the associated drug release behavior. This has been included in the limitation of this study (lines 462-464).

Reviewer 2 Report

Comments and Suggestions for Authors

In this paper, the authors have developed a new type of nanofibrous composite material through the use of the electrospinning method that heals ruptured tendons. The article is overall written well, and the conclusions are well based on the experimental data. Some improvements to the text should be made, though:

1/ The materials and methods section should be placed before the Results and the Discussion.

2/ Figure 2 should be presented more clearly.

3/ There are some inconsistencies in the Section 2.2.2. that the authors should address. The maximal load for the normal tendons looks wrong, as the error seems larger than the average value, and the values in the text don't match the values in Figure 4B. In addition, Figure 4 should be better explained in either the text or the caption, and the 4.3.2 section doesn't match this figure in terms of solely analyzing the averaged curve.

4/ Further, the authors should compare the results of their study with other similar studies in the field.

Author Response

(The authors gave the same response as above.)

Reviewer 3 Report

Comments and Suggestions for Authors

11. As showed in Figure 1, the diameter present high values of standard deviation, specifically for indomethacin loaded nanofibers.  How can the authors justify this phenomenom? This is a crucial point which could negatively affect the release kinetics of the drug, as there are nanofibers with many different diameters.

22. In the conclusions, the authors state that the prepared nanofibers provided a sustained discharge of indomethacin for at least 30 days. However, according to the results, as showed in Figure 3 (A and B), the authors affirmed that the first peak in the indomethacin release curve occurred on day 1, which corresponds almost to 100%. So, it would not be too appropriate consider the nanofiber as sustained drug delivery for indomethacin for the time frame reported in this study (30 days). Indeed, in only five days, the concentration is decreased by 1000 times. Could this condition be beneficial in the view of a translation of clinical administration? Could please the author provide literature regarding the administrated dose of the drug?

Comments on the Quality of English Language

Minor editing of English language required.

Author Response

(The authors gave the same response as above.)

Round 2

Reviewer 1 Report

Comments and Suggestions for Authors

The manuscript is significantly improved, the authors even cite now the company incaptec. The only problem is, that the technological papers need to be primarily cited, not only the company. If the company might not survive or change the website the readers might not know the topics or technology. For this reason, citing also the paper would be helpful.

Gai, M.; Kurochkin, M. A.; Li, D.; Khlebtsov, B.; Dong, L.; Tarakina, N.; Poston, R.; Gould, D. J.; Frueh, J.; Sukhorukov, G. B. In-Situ NIR-Laser Mediated Bioactive Substance Delivery to Single Cell for EGFP Expression Based on Biocompatible Microchamber-Arrays. J. Control. Release 2018, 276 (28), 84–92. https://doi.org/10.1016/j.jconrel.2018.02.044.

Author Response

Reviewer #1

The manuscript is significantly improved, the authors even cite now the company incaptec. The only problem is, that the technological papers need to be primarily cited, not only the company. If the company might not survive or change the website the readers might not know the topics or technology. For this reason, citing also the paper would be helpful.

Gai, M.; Kurochkin, M. A.; Li, D.; Khlebtsov, B.; Dong, L.; Tarakina, N.; Poston, R.; Gould, D. J.; Frueh, J.; Sukhorukov, G. B. In-Situ NIR-Laser Mediated Bioactive Substance Delivery to Single Cell for EGFP Expression Based on Biocompatible Microchamber-Arrays. J. Control. Release 2018, 276 (28), 84–92. https://doi.org/10.1016/j.jconrel.2018.02.044.

Response: Thank you for the comment. Reference 65 has been updated.

Reviewer #2

1/ The journal states it has no strict formatting requirements but that the article must contain the following sections: "Author Information, Abstract, Keywords, Introduction, Materials & Methods, Results, Conclusions, Figures and Tables with Captions, Funding Information, Author Contributions, Conflict of Interest and other Ethics Statements". Therefore, I still feel the article would read better if the authors were to place the Materials & Methods section before the Results and the Discussion.

Response: Thank you for the comment. In accordance with the journal's formatting guidelines, the section order is structured as follows: 'Introduction,' 'Results,' 'Discussion,' 'Materials and Methods,' and 'Conclusions.'

2/Figure 4B still states a different error compared to the text above the figure (4.7N vs 7.7 N). The authors should fix this and show the correct value of the error. I would recommend that the authors only show the average maximum load without error range in Figure 4B or, even better, that they display all three curves per group in the same Figure (six curves in total), given they did three tests per group. In addition, since the error is greater than the mean value, that would indicate significant negative values of the force during measurement. Can the authors further explain how that might occur during the tensile test?

Response: Thank you for the comment. The typo „4.0 ± 7.7“ has been revised to „4.0 ± 4.7“ (line 192). Significant error bars in Figure 4B have resulted in negative values from a mathematical perspective. This issue is primarily attributable to the limited number of animals involved in the in vivo study. We have acknowledged this limitation in the manuscript (lines 460-461).

Reviewer 2 Report

Comments and Suggestions for Authors

I would like to thank the authors for improving their manuscript based on the previous feedback. Please see my feedback on the revised version below:

1/ The journal states it has no strict formatting requirements but that the article must contain the following sections: "Author Information, Abstract, Keywords, Introduction, Materials & Methods, Results, Conclusions, Figures and Tables with Captions, Funding Information, Author Contributions, Conflict of Interest and other Ethics Statements". Therefore, I still feel the article would read better if the authors were to place the Materials & Methods section before the Results and the Discussion.

2/Figure 4B still states a different error compared to the text above the figure (4.7N vs 7.7 N). The authors should fix this and show the correct value of the error. I would recommend that the authors only show the average maximum load without error range in Figure 4B or, even better, that they display all three curves per group in the same Figure (six curves in total), given they did three tests per group. In addition, since the error is greater than the mean value, that would indicate significant negative values of the force during measurement. Can the authors further explain how that might occur during the tensile test?

Author Response

(The authors gave the same response as above.)
